# Evaluation of Nutritional Quality and Sensory Parameters of Meat from Mallard and Four Species of Wild Goose

**DOI:** 10.3390/foods11162486

**Published:** 2022-08-17

**Authors:** Pär Söderquist, Camilla Olsson, Karina Birch, Viktoria Olsson

**Affiliations:** Faculty of Natural Science, Kristianstad University, 29188 Kristianstad, Sweden

**Keywords:** animal-based protein, sustainable food, food and health, sensory evaluation, nutrition, barnacle goose, bean goose, Canada goose, greylag goose, Anas, anser, Branta

## Abstract

Future challenges concerning protein supply for food and feed include the management of all currently available resources. In Sweden, wildfowl are hunted for several reasons, one of which is to protect growing crops. In this study, meat from wild geese and mallard was evaluated with respect to its quality and sensory parameters. The most pronounced sensory differences were between meat from the barnacle goose and the Canada goose and between meat from mallards that were farmed and born wild. This study also provides measurements of values for the nutritional and heavy metal contents of the meat from these wildfowl species in order to elucidate their possible use as modern foods.

## 1. Introduction

The increasing global population, which is expected to exceed 9 billion people by 2050, makes food and nutrition security a central global issue and, ensuring access to affordable healthy diets for all, urgent. The global food system is currently failing to meet nutritional needs and there are growing concerns for health in relation to both under and over consumption [1], as well as micronutrient deficiency with iron being a special concern. Globally, the availability of high-quality protein is restricted [2] and the unit costs of protein are generally higher than those of carbohydrates or fats, driving nutritional inequalities. In addition to producing enough calories to feed a growing global population, the food system must also provide a diversity of food types that nurture human health and support environmental sustainability [3]. Currently, 90% of high-quality animal protein derives from only a small array of animal species meaning that diversification would be beneficial. The resource footprints of animal-based products are large compared to those of plant-based alternatives due to the fact that inefficient feed conversion in the production of livestock involves sizeable losses of energy and digestible protein [2,3]. Land and water use, greenhouse gas emissions, and concerns related to animal welfare are all examples of important challenges that are generally associated with the provision of animal proteins in the diet. Currently, in Sweden, the national degree of self-sufficiency regarding food is another factor that has risen up on the agenda, with more domestically produced foodstuff being called for. There is, therefore, an urgent need to optimize the use of locally available animal protein resources and, in this respect, game meat may play a role. Even though game meat is generally considered to have environmental and ethical advantages over industrially produced meat, as well as numerous culinary virtues, it has remained rather unfamiliar to many consumers in Sweden; this is especially the case with regard to wildfowl. Meat derived from wild animals and birds is synonymous with heterogenous characteristics and considerable variation in chemical composition, with differences in factors such as age, sex, body condition, sexual status and season affecting a wide range of meat properties [4,5]. Thus, in the context of utilizing local resources, there are still several sensory and nutritional aspects of game meat, including that of wildfowl, worthy of further exploration.

Many of the wild goose populations in Europe have increased rapidly, and most population numbers are now higher than ever due to increased survival and reproductive success as a result of changes in the agricultural landscape, climate, and hunting [6,7,8]. Beyond this rapid growth, several species have also extended their range and now spend less time in their natural habitats of wetlands and grasslands in favor of increasingly exploiting agricultural areas for feeding. Changes in the number and distribution of geese have led to conflicts with human interests, especially within agriculture where grazing geese cause damage to growing crops [9]. Various methods are available for preventing and managing such damage and the resulting conflicts related to geese. Hunting and derogation shooting are often used in combination with other methods in attempts to regulate population size or to alter the movements of the geese away from sensitive crops or other areas where conflict with human interest arises. The harvest of wild geese in Sweden currently exceeds 50,000 birds a year and concerns three species, namely the bean goose (Anser fabalis), Canada goose (Branta canadensis), and greylag goose (Anser anser), with the addition of an increasing number of barnacle geese (Branta leucopsis) which is being reduced by means of derogation hunting [10]. Europe has a long-standing tradition of goose hunting, but also of rearing farmed geese for meat consumption. Originating from the wild greylag goose, selective breeding for intensive meat production has resulted in numerous varieties of the domestic goose (Anser anser domesticus) which now dominates the commercial market for goose meat [11]. Despite an increasing demand for poultry meat nationwide, domestic geese constitute a humble share of the total poultry production in Sweden, with less than 10,000 birds (<0.01%) slaughtered during 2021 [12].

Another common wildfowl, the mallard, (*Anas platyrhynchos*), is the most well-known, widespread, and numerous duck species with an estimated global population of more than 19 million breeding individuals [13]. It is also one of the world’s most important game species with an estimated annual harvest of approximately 4 million mallards in Europe and Russia [14,15]. The number of mallards shot in Sweden has increased from 75,000 to almost 300,000 birds per year in only 25 years and the trend is still upward [16]. The European population of mallards has been declining in some parts over the past decades [17,18]; however, the Nordic populations have shown a stable or positive trend during the same period [17]. In order to maintain a high population for hunting, massive management efforts have been undertaken, e.g., wetland restorations and supplementary restocking of wild populations with farmed mallards. Restocking mallard populations with individuals raised in captivity is a practice dating back to the early 1900s in North America [19]. In Europe, the practice became more common in the 1970s and, although hard to estimate, more than 5.5 million farmed mallards are released in Europe per year for hunting purposes [20,21], at least 250,000 of which in Sweden [22]. The general method for rearing mallards for release starts with mallard eggs being produced in breeding facilities by birds originating from wild-trapped and domestic lines. Between May and July, hatched ducklings aged two-three weeks are released into wetlands where hunting will take place during the fall. After release, the ducklings are fed, often with barley (Hordeum vulgare), until the end of the season. This practice is similar in most countries releasing farmed mallards for hunting purposes. Although the rearing and release of mallards are mainly for hunting purposes, mallard is frequently offered on the menu at restaurants, and it is estimated that more than 681 million mallards per year are farmed worldwide for food consumption [23]. The physical characteristics and nutritional value of the meat have remained poorly investigated and considering the substantial variations in rearing conditions between farmed and wild mallard, a comparison of the meat qualities is much needed.

The recent increase in the harvest of geese and mallards provides an opportunity to meet future challenges concerning sustainable alternative protein sources through the management of locally available resources. Considering that the harvest of geese now includes several species, and that mallards consist of both “truly wild” and farmed-released, a possible differentiation in meat characteristics could provide new culinary experiences for consumers. Knowledge about the nutritional value and sensory attributes of meat has long been recognized as an important factor that influences consumer preference, attitudes and behavior towards the consumption of game meat products [24]. The objectives of this paper were to describe game meat from local wildfowl concerning meat quality parameters, such as nutritional and heavy metal content, as well as sensory attributes. The explicit aims were to answer the questions stated below concerning four locally abundant goose species (barnacle goose, bean goose, Canada goose and greylag goose), and mallard. Mallard was further explored regarding sex and type (farmed and wild).

Research questions
What are the physical characteristics and nutritional values of the meat from the studied wildfowl species?To what extent are traces of heavy metals present in meat from the studied wildfowl species?What are the sensory characteristics of the meat from different goose species?Is it possible to discriminate between meat from mallards regarding gender and type?

## 2. Materials and Methods

### 2.1. Sampling

Meat from four goose species—barnacle goose, bean goose, Canada goose, and greylag goose (six individuals per species, respectively)—were obtained from a local butcher’s supplying game meat in Skåne county, southern Sweden, during the fall of 2018.

Meat from 12 farmed (six females and six males) and 12 presumed wild (six females and six males) mallards were obtained from hunts in Småland county, southern Sweden, during the fall of 2018. All farmed mallards had been fitted with a conventional steel ring provided by the Swedish Museum of Natural History prior to release in 2017–2018. This ensured that truly farmed mallards were used in the study. Mallards without rings were considered to be wild.

The carcasses were plucked, dressed, dissected, vacuum packed, and frozen at −18 °C until further preparation. Only breast muscles (musculus pectoralis major) were used in this study.

### 2.2. Instrumental Analyses

Thawed breast muscles were color-determined instrumentally, using a CM700d Chroma Meter Measuring Head (Konica Minolta, Tokyo, Japan) and weighed (raw weight) using a digital scale (Mettler-Toledo, Columbus, OH, USA). To determine cooking loss (difference between raw weight and weight after cooking), samples were weighed after being cooked in sous-vide manner and then cooled in preparation for sensory analyses (see Section 2.4).

### 2.3. Chemical Analyses

In preparation for the chemical analysis of nutritional content (carbohydrates, protein, lipids, salt, iron, energy and water contents, full fatty-acid composition) and heavy metal content (mercury (Hg), lead (Pb), arsenic (As), and cadmium (Cd)), the samples were semi-thawed and homogenized (Robot Coupe Blixer 4, Robot Coupe, Montceau-les-Mines, France). Samples from all individuals of the respective goose species and mallards (four groups) were homogenized within each group for all analyses, except for the analysis regarding iron and heavy metal content where individual mallards were considered separately. All samples were sent to an accredited, commercial laboratory (ALS Scandinavia AB, Luleå, Sweden) for analysis. The nutritional content was analyzed using NMR, GC-FID, IC-ECD, AOAC 985.29, Dumas, ICP-AES and gravimetry, fatty acid composition using GC-FID and heavy metals and iron using ICP-SFMS.

### 2.4. Sensory Analyses

In preparation for sensory analysis, individual breast muscles were thawed, vacuum packed, and cooked in a sous-vide manner to an internal temperature of 59 ± 1 °C. After cooking, the samples were cooled and cut into approximately 7 mm thick slices and placed in coded petri dishes.

Sensory evaluation was carried out in a sensory laboratory at Kristianstad University designed according to ISO 8589. The panel consisted of nine assessors, selected and trained according to the following guidelines: ISO 3972:2011, and ISO 8586:2014. Product evaluations were performed individually by the assessors in isolated booths. Samples were coded with three-digit codes and served in a randomized order. The serving temperature was adjusted to eating temperature by incubating the samples at 70 °C for 10 min prior to serving. Panelists were instructed to rinse their mouths with lukewarm water, either still or carbonated, after each sample and were also provided with wheat crackers for further palate cleansing.

For the mallards, sensory discrimination between type (farmed and wild) and sex was tested using a triangle test (ISO 6658:2017). The panelists were presented with three samples, of which two were of the same type and one was different. The panelists were obliged to state which sample was different and the results then indicate whether a detectable difference exists between the two samples.

Sensory analysis of the four goose species was performed using Quantitative Descriptive Analysis (QDA; ISO 6658:2017). Across two training sessions, lasting approximately 2 h each, the panel developed descriptions of the perceived sensory attributes of the products, generating a set of attributes and developing a consensus regarding the evaluation of each specific attribute (Table 1). During training, reference materials were used for selected attributes, such as total color intensity, odor and flavor of liver and iron. The panelists then evaluated the perceived intensities on a continuous 100 mm line-scale which labeled “low intensity” at 10 mm and “high intensity” at 90 mm using the software EyeQuestion (v4.11.62, Elst, The Netherlands). The panelists evaluated triplicates of each product during one evaluation session lasting 60 min.

### 2.5. Statistical Analyses

A one-way ANOVA in IBM SPSS Statistics version 24 was used to test for differences between the four goose species regarding weight before and after cooking, L*, a*, and b*. Sensory data from the QDA test were subjected to a two-way ANOVA in EyeOpenR, with samples, panelists, and replicates as fixed effects. Tukey’s post-hoc tests were used to contrast between species with significant outcomes. Independent sample t-tests were used to test for differences between mallard type (wild and farmed) and between sex within each respective group, regarding weight before and after cooking, L*, a*, b*, iron content, and heavy metals (Hg, and Pb). Levene’s test for homogeneity of variances and Levene’s test for equality of variances were used for all ANOVAs and t-tests, respectively. The statistical significance of the triangle test was analyzed based on binomial distribution in accordance with Lawless and Heymann [25]. A significance level of 0.05 was used in all tests.

### 2.6. Ethical Considerations

No ethical approval was needed to conduct the study. Mallards and geese were collected from hunts and were not specifically shot or reared for this study. The ringing of mallards prior to release was performed by bird ringers licensed by the Swedish Bird Ringing Centre. All panelists performing sensory evaluations signed a written consent regarding participation in the study and use of the results.

## 3. Results

### 3.1. Nutritional Content

Meat from all four species of goose and from all four groups of mallards had an energy content of between 124–139 kcal per 100 g (Table 2). The content of protein varied between 23.4–24.6 g per 100 g meat (Table 2). Goose meat contained less than 3 g of fat per 100 g, except for that of Canada goose which contained 4.17 g (Table 3). Meat from wild mallards contained less than 3 g of fat per 100 g and farmed mallard meat between 3.3–3.5 g per 100 g. Meat from all four species of goose and all four groups of mallards had a low content of saturated fatty acids. The bean goose, barnacle goose and the Canada goose all contained 40 mg of eicosapentaenoic acid and docosahexaenoic acid per 100 g (Table 3).

Iron content was significantly higher in meat from wild mallards compared to farmed for both males and females (males: *p* = 0.012, t = 3.048, df = 10; females: *p* = 0.03, t = 2.531, df = 10). However, no significant differences in iron content were found between males and females within respective type (*p* > 0.177).

Complete nutritional information and a selection of fatty acids presented as grams per 100 g of fat can be found in Table 2 and Table 3 respectively. A full fatty acid profile is provided as Appendix A.

### 3.2. Heavy Metal Content

No significant differences were found in mercury content between wild and farmed mallards (*p* > 0.126), nor between males and females within respective type (*p* > 0.493) (Table 4). No significant differences could be found in lead content between wild and farmed mallards (*p* > 0.162) nor between males and females within respective type (*p* > 0.213). Meat from all groups was also analyzed for arsenic and cadmium content and all groups were found to contain less than 0.01 mg/kg of arsenic and less than 0.005 mg/kg of cadmium (Table 4).

### 3.3. Physical Characteristics

A significant difference was found in both raw weight (*p* < 0.001, F = 20.758, df = 3) and cooked weight (*p* < 0.001, F = 18.599, df = 3) between the four species of goose, with meat from barnacle goose weighing less than meat from the other species (raw weight: *p* < 0.003; cooked weight: *p* < 0.006) (Table 5). Both raw and cooked weights were significantly higher in Canada goose than in bean goose (*p* < 0.022). In addition, cooking loss differed significantly between goose species (*p* = 0.006, F = 6.224, df = 3), with barnacle goose losing more weight than both bean goose (*p* = 0.019) and Canada goose (*p* = 0.006) (Table 5).

For mallards, raw and cooked weights were significantly higher in wild males compared to farmed males (raw weight: *p* = 0.032, t = 2.485, df = 10; cooked weight: *p* = 0.024, t = 2.656, df = 10). For the wild mallard type, males had significantly higher raw and cooked weights than females (raw weight: *p* = 0.003, t = 3.874, df = 10; cooking weight: *p* = 0.003, t = 3.847, df = 10). No other significant differences were found for raw or cooked weights, respectively (*p* > 0.078). There were no significant differences in cooking loss for any of the groups (*p* > 0.133).

No significant differences in instrumentally determined meat color were found between the four goose species (*p* > 0.066). The same was also true for mallards (*p* > 0.123).

### 3.4. Sensory Evaluation

In the sensory evaluation, meat from the bean goose and the greylag goose was perceived as having a more intense total meat and iron odor than meat from the Canada goose (*p* ≤ 0.05) (Figure 1). Barnacle and Canada goose meat significantly differed concerning all attributes related to appearance, i.e., total color intensity, glossiness, coarseness of fiber structure (*p* ≤ 0.05). The assessment by the sensory panel of fiber structure found that the meat structures of bean goose and greylag goose were visually similar to that of Canada goose. Regarding textural properties, the barnacle goose meat was found to be more tender (*p* ≤ 0.001) and had lower cutting resistance (*p* ≤ 0.001) than the other species. Furthermore, barnacle goose meat was also perceived as crumblier than meat from the bean goose and Canada goose, while there was no significant difference in juiciness between any of the four goose species studied. No significant differences were found among any of the species regarding taste or flavor of their respective meat. No differences could be found between meat from the bean goose and the greylag goose in any of the sensory characteristics.

The sensory discrimination tests showed that it is possible to discriminate between a wild and a farmed female mallard (*p* = 0.043), as well as between a wild and a farmed male mallard (*p* = 0.043). No discrimination was however possible between a male and a female of the same type (*p* = 0.391 in both cases).

## 4. Discussion

The results show that, on a general level, the type of rearing and species can affect meat quality parameters, such as nutritional contents, as well as sensory attributes in wildfowl. Game meat is well known for having heterogenous characteristics and high variation in chemical composition [4,5], but the extent of variation within and between species has not been extensively studied regarding meat from goose and mallard. Although the geese in this study originated from the same harvest area with similar conditions regarding climate and vegetation, the life-history of the birds is likely to vary, on both a short-term and long-term basis. The potential causes of the differences observed in meat characteristics of the different goose species and mallards of different sexes and rearing conditions, are not within the scope of this study and can only be speculated. However, based on the extensive research available on the impact of diet on the physical properties and sensory characteristics of commercially farmed domesticated geese [26], it would not be controversial to suggest that the potential difference in diet among goose species and in wild versus farmed mallards could be an important factor. Forage feeding, as compared to more intensive feeding regimes, has long been known to affect growth performance and meat quality in for example cattle [27]. A study of Muscovy ducklings indicated how diet may influence sensory properties, showing that a plant protein diet, as compared to fish meal-containing diets, significantly improved flavor, tenderness, juiciness and general acceptability of the meat [28]. Farmed mallards rely heavily on supplemental feeding, such as barley, compared to wild mallards who rely more on seeds, buds and invertebrates [29,30,31]. This difference in diet may explain the result of the discrimination test. The wild mallards in our study had heavier breast muscles and a lower fat content than their farmed conspecifics, and similar differences in meat characteristics have been shown in other studies comparing waterfowl subjected to different feeding regimes (grain fed versus forage fed) [32].

Although most goose populations in Europe have started to increasingly utilize agricultural land for feeding rather than natural grasslands and wetlands, there are variations in the extent to which they do so, the crops they select, and how much time of the year they spend in different habitat types [9]. In southern Sweden, most of the barnacle and bean geese that are present on agricultural fields in the fall are staging or wintering geese who return to breeding areas further north in the spring [33]. In contrast, the majority of the greylag and Canada geese in southernmost Sweden are resident, and therefore winter and breed locally. Prior to episodes of high energy expenditure, such as migration, reproduction or maintaining body condition throughout wintering, the geese must balance their energy and nutrient budget and acquire appropriate body stores in the form of fat or glycogen for fuel [34]. Depending on their current state, as well as the type of energy expenditure to be endured, geese are highly selective foragers and adapt their feeding strategy accordingly with differences in diet as a result. Moreover, there are local variations in terms of the availability of certain foods and habitats to consider, both on a spatial and a temporal scale, adding to the variation in diet amongst wild geese. In addition to potential differences in movement and diet, the goose species included in this study vary in both size and genetic relatedness. The greylag goose and bean goose, both descendants from the Anser genus, are similar in both size and appearance, whereas the Canada goose and barnacle goose from the Branta genus represent the largest and smallest of the four studied species. Muscle fiber composition is affected by several factors, such as muscle type, activity, and age of the animal, and plays a key role in meat quantity and quality. Poultry meat with different types of muscle fibers display differences in color, tenderness, and water-holding capacity [35]. However, according to Weng [35], the muscle fiber characteristics of domestic geese of different ages, and the effect of muscle fibers on goose meat quality, have not been thoroughly documented. The sensory data in our study indicate differences in the meat appearance of different wild goose species, especially concerning barnacle and Canada goose. Meat from the Canada goose was perceived to have a more intense total color intensity and glossiness, and a significantly coarser fiber structure (*p* ≤ 0.05) compared to that of the barnacle goose. As documented by for example Weng [35] the muscle fiber composition can correlate to instrumental shear force; in this study meat from the barnacle goose, with a finer fiber structure (visually determined by the analytical panel), was perceived as more tender (*p* ≤ 0.001) and with lower cutting resistance (*p* ≤ 0.001) than meat from the Canada goose. In this context, the lower water holding capacity of meat from the barnacle goose, compared to both the bean goose and Canada goose, is worth noting. In future research on this matter, histochemical analysis for estimation of muscle fiber characteristics in different wild goose species would be relevant, and in that case red (thigh) muscles could be of interest to include.

Although mallards have been released for hunting purposes on a large scale for almost 50 years in Sweden, this is not particularly well known among the general public. It is probably also not widely known among restaurants which serve “wild duck” that these mallards are largely bred in captivity. On the other hand, it is known that meat from mallard has a high meat quality and that it is highly appreciated in restaurants [36]. The mallard is omnivorous and opportunistic, and their diet, which changes by season and sex, consists mostly of invertebrates and seeds [29,30,31]. Behavioral differences between wild and farmed mallards have long been known. Already in the 1930s, studies showed that farmed mallards had shorter migrations than their wild conspecifics [19]. Later studies corroborated shorter migrations and movements of farmed mallards [37,38,39], but also lower survival [21,40,41]. Morphological differences have also been found between wild and farmed mallards. Released farmed mallards have long been known to be heavier and have a stockier body compared to their wild counterparts [19,31]. It is therefore interesting that the breast muscles of the farmed mallards had both lower raw and cooked weights compared to their wild conspecifics. The more sedentary lifestyle of farmed mallards, together with supplemental food based on grains, may lead to heavier bodies but smaller flight (breast) muscles. The bill of a mallard is a complex structure that has evolved to sieve out small food particles from the water [42]. However, as farmed mallards are given supplementary feed containing larger food particles, such as barley, the selective pressure for dense bill lamellae to sieve out smaller food particles has been reduced, leading to a decrease in lamellar density in farmed mallard bills [43]. The bills of released farmed mallards also appear wider, higher, and shorter, i.e., more goose-like, than those of wild mallards [31,43]. Finally, by studying the genetics of wild and farmed mallards in Europe, it is clear that farmed mallards are genetically different to their wild conspecifics [21,44,45].

Although the underlying causes of the differences seen in meat characteristics between farmed and wild mallards, and between geese of different species, remain unknown, behavioral, genetic and morphological differences are all plausible explanations. It is interesting that the trained sensory panelists could discriminate between wild and farmed mallards but not between male and female mallards. Future studies on the details of the differences in meat characteristics between mallards having different rearing conditions and how these affect the acceptance of meat from mallard would be interesting in relation to an increase in consumption of local wildfowl meat.

Meat from all four goose species and from all four groups of mallard contains valuable nutrients, is rich in protein (more than 20% of its energy content) and contains low amounts of saturated fatty acids. Although only present in relatively low concentrations, the long-chained omega-3 fatty acids, almost exclusively found in marine food sources, are also found in three of the four goose species. Previous studies have shown that game meat contains omega-3 fatty acids but not in sufficiently high concentrations to be regarded as an adequate replacement for marine food sources (mainly fish) in the diet. In this regard, meat from wild geese would seem to constitute a viable option.

Iron exists in the form of haem iron or non-haem iron, where haem iron is commonly found in meat products and is absorbed more efficiently when consumed compared to non-haem iron, which is mainly found in plants and grains [46]. The meat from mallard and all species of goose contained high levels of iron, even when compared to other food sources normally cited as especially rich in haem iron, such as moose. Our results further suggest a slightly higher iron content in meat from female mallards; similar patterns have been found in previous studies on beef where meat from cows contained higher levels of iron than meat from bulls [47].

One objective of this study was to examine for traces of heavy metals in the studied wildfowl. The levels of arsenic (As) and cadmium (Cd) were lower than the detectable limits, and levels of mercury (Hg) were only detectable in wild mallards, although still below the recommended threshold for, e.g., fish products. Traces of lead (Pb) were, however, found in all groups and were above the recommended threshold for barnacle goose, greylag goose, and wild female mallards. The results indicate extremely high concentrations of lead in the sample from the barnacle geese, which may pose a health risk if consumed regularly over long periods of time. Lead contamination in the meat could arise from active hunting with lead ammunition (including damage shooting), but also through the birds ingesting lead pellets while foraging in wetlands previously used for hunting [48]. Although the use of lead ammunition in wetland areas has been banned for decades, most geese are hunted on agricultural fields and not wetlands, especially the barnacle geese which are only hunted as a means of derogation shooting on growing crops. The sample size in this study is very limited, and therefore the lead content should be interpreted with caution since there could be great variation between individuals of the same species. However, the results call for further studies on this matter.

A factor that was not included in the scope of this study, but which should be discussed, is the age of the birds. The age of an animal has been shown to have significant effects on the properties of the meat, e.g., Weng, [35]; however, since it is not possible to accurately determine the age of adult mallards or geese from wild populations through ocular inspection, age could only be determined for the farmed mallards and was therefore not included in the analyses. Another uncertainty is the fact that it is impossible to know if the mallards that were treated as “wild” in the study were truly wild or in fact unmarked individuals from farmed populations.

Nonetheless, the study points to significant effects of both species and rearing types for the wildfowl studied. An understanding of these variations could lead to better use being made of this unique resource in terms of, for example, its nutritional contribution to the diet and to culinary practice.

## 5. Conclusions

Meat from the studied wildfowl species is nutritious, characterized by high protein and low total fat and saturated fatty acid contents. It is also relatively rich in valuable nutrients, such as eicosapentaenoic acid, docosahexaenoic fatty acids and iron. The levels of arsenic, cadmium and mercury in the meat were lower than the recommended thresholds and/or detectable limits. Traces of lead (Pb) were, however, found in all the studied wild fowl species and were above the recommended threshold for barnacle goose, greylag goose, and wild female mallards.

The weight of the breast muscle varied between the goose species studied and meat from the barnacle goose exhibited lower water holding capacity, measured as higher cooking loss, compared to the other geese. The breast muscles of wild male mallards were heavier than the farmed. Overall, no significant differences in instrumentally determined meat color were found.

In the sensory evaluation, barnacle goose meat stood out in the assessment of appearance and texture. The analytical panel perceived this meat as having a less intense color, finer fiber structure, and as more tender when compared to meat from the Canada goose. However, no significant sensory differences were perceived regarding odor and flavor when comparing the four goose species. The discrimination tests showed that it is possible to discriminate between meat from wild and farmed mallards but not between the meat from males and females. To conclude, these exploratory data can contribute to better use of a unique protein resource in the form of meat from wild fowl.

## Figures and Tables

**Figure 1 foods-11-02486-f001:**
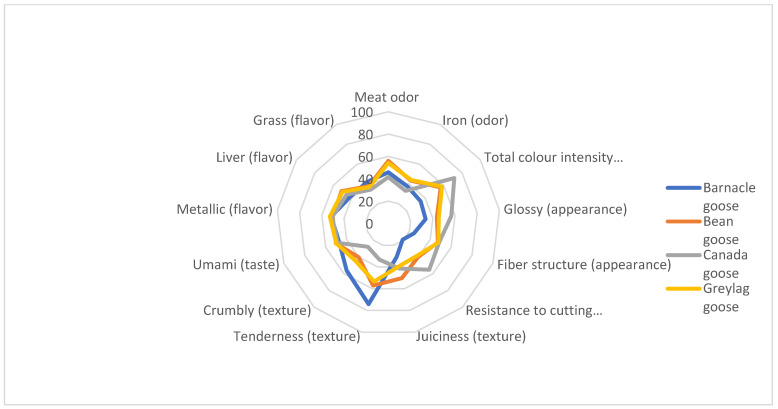
Sensory characteristics of meat from barnacle goose, bean goose, Canada goose and greylag goose.

**Table 1 foods-11-02486-t001:** Sensory attributes and definitions used in the sensory analysis of the four goose species.

Attribute	Type	Definition
Total color intensity	Appearance	The intensity of the red color (low-high)
Glossy	Appearance	Glossiness
Fiber structure	Appearance	0 for fine fibers and 100 for coarse
Resistance to cutting	Texture	Resistance when using cutlery in a standardized manner, from low to high resistance
Juiciness	Texture	Use molar teeth. Assess texture in mouth with 1-2-3 chewings. Assess juiciness (low-high) at first bite.
Tenderness	Texture	Use molar teeth. Assess texture in mouth with 1-2-3 chewings. Assess tenderness (low-high) at second bite.
Crumbly	Texture	Use molar teeth. Assess texture in mouth with 1-2-3 chewings. Assess crumbliness/grainy texture (low-high) at third bite.
Meat odor	Odor	Rich meat odor, broth odor
Iron	Odor	Metallic odor from cooked liver
Umami	Taste	Basic taste
Metallic	Flavor	Flavor of iron and/or blood
Liver	Flavor	Flavor of cooked liver
Grass	Flavor	Green, a chlorophyllic sweetness.

**Table 2 foods-11-02486-t002:** Nutritional information per 100 g meat for all four species of goose, as well as farmed and wild mallards of both sexes. The values represent mean values based on pooled samples of six breast muscles from each group, except for iron content in mallards that was analysed separately for each individual, giving mean values ± sd. Significant differences in iron content are indicated with uppercase letters ^a^ and ^b^.

	Barnacle Goose	Bean Goose	Canada Goose	Greylag Goose	Farmed Male Mallard	Farmed Female Mallard	Wild Male Mallard	Wild Female Mallard
Energy (kJ/kcal)	517/124	525/125	580/139	531/127	539/129	525/125	517/124	521/125
Carbohydrate (g)	0.7	<0.30	1.22	1.02	<0.30	<0.30	<0.30	0.71
Glucose (g)	<0.05	0.11	0.35	0.14	<0.05	0.18	0.15	0.14
Protein (g)	23.6	24.3	23.8	24.6	24.2	23.4	24.3	24.4
Fat (g)	2.8	2.9	4.17	2.58	3.47	3.32	2.7	2.57
Saturated fat (g)	0.86	0.82	1.24	0.74	1.2	1.18	0.92	0.87
Monounsaturated fat (g)	0.77	1.21	1.59	1.06	1.28	1.16	0.87	0.76
Polyunsaturated fat (g)	1.05	0.74	1.15	0.67	0.83	0.83	0.8	0.82
Omega-3 fatty acids, total (g)	0.46	0.24	0.41	0.12	<0.10	<0.10	<0.10	<0.10
Omega-6 fatty acids, total (g)	0.59	0.5	0.75	0.54	0.83	0.83	0.74	0.8
Salt (NaCl) (g)	0.15	0.14	0.15	0.13	0.14	0.13	0.15	0.15
Iron (mg)	5.65	7.03	6.85	6.98	4.65 ± 0.55 ^a^	4.88 ± 0.72 ^a^	5.46 ± 0.35 ^b^	5.97 ± 0.77 ^b^
Water (g)	71.5	70.7	69.6	70.6	71.3	71.2	71.2	71.1

**Table 3 foods-11-02486-t003:** A selection of relevant fatty acids in grams per 100 g of fat for all four species of goose, as well as for farmed and wild mallards of both sexes. The values represent mean values based on pooled samples of six breast muscles from each group.

	Barnacle Goose	Bean Goose	Canada Goose	Greylag Goose	Farmed Male Mallard	Farmed Female Mallard	Wild Male Mallard	Wild Female Mallard
C16:0 Palmitic acid	16.4	17.2	18.6	18.0	21.1	21.7	18.8	18.3
C16:1 Palmitoleic acid	2.09	2.29	2.3	3.32	1.7	1.48	1.94	1.71
C18:0 Stearic acid	14.2	11.2	11.1	10.8	13.5	14.0	15.2	15.7
C18:1n9c Oleic acid	23.8	37.1	34,0	35.3	32.5	30.9	28.1	25.0
C18:2n6c Linoleic acid	14.2	11.1	13.1	14.2	14.0	13.6	14.8	16.9
C18:3n3 Linolenic acid	14.6	6.51	7.96	4.74	<0.05	<0.05	1.04	1.03
C20:4n6 Arachidonic acid	6.77	6.1	4.81	6.93	10.0	11.4	12.6	14.1
C20:5n3 Cis-eicosapentaenoic acid	1.92	0.94	1.02	<0.05	<0.05	<0.05	1.02	<0.05
C22:6n3 Cis-docosahexaenoic acid	<0.05	0.7	0.76	<0.05	<0.05	<0.05	<0.05	<0.05
Omega-3 fatty acids, total	16.6	8.16	9.74	4.74	<0.20	<0.20	2.06	1.03
Omega-6 fatty acids, total	21.0	17.2	17.9	21.1	24.0	25.0	27.4	31.0

**Table 4 foods-11-02486-t004:** Heavy metal content. Mean values in mg/kg (±sd) for mallards that were analysed (individually) for mercury (Hg), lead (Pb), arsenic (As), and cadmium (Cd). For each goose species, six breast muscles were pooled and no deviation from the mean could be calculated. No significant differences between any groups were found (*p* > 0.126). As no limit values for wildfowl exist, limit values for some various food for each heavy metal are included as reference.

	Barnacle Goose (*n* = 6)	Bean Goose (*n* = 5)	Canada Goose (*n* = 5)	Greylag Goose (*n* = 5)	Farmed Female Mallard (*n* = 5)	Farmed Male Mallard (*n* = 6)	Wild Female Mallard (*n* = 6)	Wild Male Mallard (*n* = 6)
Hg *	<0.01	<0.01	<0.01	<0.01	0.01 ± 0.0	0.01 ± 0.00	0.019 ± 0.019	0.028 ± 0.025
Pb **	1.85	0.02	0.089	0.148	0.035 ± 0.027	0.03 ± 0.012	0.31 ± 0.42	0.069 ± 0.090
As ***	<0.01	<0.01	<0.01	<0.01	<0.01	<0.01	<0.01	<0.01
Cd ****	<0.005	<0.005	<0.005	<0.005	<0.005	<0.005	<0.005	<0.005

* 0.5–1.0 mg/kg = Limit value for fish/seafood, ** 0.1 mg/kg = Limit value for beef, mutton, pork and poultry, *** 0.1–0.3 mg/kg = Limit value for rice cakes, **** 0.050 mg/kg Limit value for beef, mutton, pork and poultry.

**Table 5 foods-11-02486-t005:** Physical characteristics of meat from goose and mallard. Mean values (±sd) for raw weight (g), cooked weight (g), and cooking loss (%), together with the number of individual breast muscle samples (*n*) in all groups. Mean values (±sd) for lightness (L*), redness (a*), and yellowness (b*) are also given for each group. Significant differences between groups for each characteristic are indicated with uppercase letters ^a^ to ^s^.

	Barnacle Goose	Bean Goose	Canada Goose	Greylag Goose	Farmed Male Mallard	Farmed Female Mallard	Wild Male Mallard	Wild Female Mallard
*n*	6	5	5	5	6	6	6	6
Raw weight (g)	130.4 ± 18.7 ^a^	227.9 ± 26.5 ^b^	308.2 ± 61.7 ^c^	248.0 ± 35.6 ^b,c^	79.7 ± 11.7 ^i^	67.4 ± 9.9 ^i^	94.6 ± 8.8 ^j^	70.1 ± 12.7 ^i^
Cooking weight (g)	110.9 ± 17.3 ^d^	198.9 ± 23.5 ^e^	272.2 ± 58.0 ^f^	223.9 ± 21.8 ^e,f^	67.1 ± 9.9 ^k^	58.3 ± 9.1 ^k^	81.6 ± 9.0 ^l^	60.5 ± 10.0 ^k^
cooking loss (%)	17.1 ± 1.8 ^g^	12.7 ± 2.3 ^h^	12.0 ± 1.7 ^h^	14.5 ± 2.5 ^g,h^	15.7 ± 2.9 ^m^	13.6 ± 1.3 ^m^	13.7 ± 4.2 ^m^	13.5 ± 2.6 ^m^
L*	33.3 ± 3.0 ^n^	33.9 ± 2.2 ^n^	30.5 ± 2.6 ^n^	31.9 ± 2.0 ^n^	38.2 ± 8.8 ^q^	37.7 ± 3.5 ^q^	38.7 ± 3.0 ^q^	36.0 ± 2.5 ^q^
a*	12.2 ± 2.0	9.3 ± 1.2 ^o^	11.2 ± 2.0 ^o^	9.9 ± 2.3 ^o^	6.0 ± 1.3 ^r^	5.2 ± 1.5 ^r^	4.9 ± 2.1 ^r^	5.8 ± 1.4 ^r^
b*	8.6 ± 1.4 ^p^	6.3 ± 1.7 ^p^	7.8 ± 1.6 ^p^	6.9 ± 1.5 ^p^	2.6 ± 1.0 ^s^	1.9 ± 1.0 ^s^	2.4 ± 1.2 ^s^	1.6 ± 0.6 ^s^

## Data Availability

The data presented in this study are available on request from the corresponding author.

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
