# Peer review of "Evaluation of Nutritional Quality and Sensory Parameters of Meat from Mallard and Four Species of Wild Goose"

_foods, 2022, doi:10.3390/foods11162486_

Round 1

Reviewer 1 Report

Remarks

  1. Goose meat contained high values of lead is it recommended to use it for food (Table 4 data)?
  2. Is it unclear what the norms are in the footnotes to Table 4? Why are the norms for fish mentioned? Rice cakes?(line 234, 237).

3.Line 246: Is it necessary to specify the methodology for assessing the water-binding capacity? in what kind of meat was determined (boiled or raw?) clarification is required on which characteristics this affects to clarify, or to present in Chapter 5.

  1. Line 255-257: Since the muscles of the chest (musculus pectoralis major) were used in this study, it is possible that differences will be found in the red (dark) meat of goose and mallard, to which the thigh meat belongs. The muscles of the thigh contain mainly fibers of a rapid type of oxidation (glycolytic fibers), unlike the muscles of the chest. Therefore, the choice of muscle type for research is not entirely clear. The main loads fall on the motor (wing, hip) muscles and the oxidation processes take place in these muscles more intensively. Therefore, it is more expedient to choose red (thigh) muscles for study. This is better suited for assessing the sensory profile, water binding capacity and fatty acid fraction of farm and wild poultry. Therefore, the authors did not find significant differences in the selected muscle type.
  2. Chapter 3.4 (recommended to authors): the above conclusions about the sensory characteristics of meat should be linked to the habitat halo, feeding diets or breed characteristics when discussing the results of this chapter (in the discussion given in lines 284-297, and linked to the data specified in Chapter 3.4)
  3. Clarify lines 278 and 281: there is a feeling of contradiction between the conclusions of the first and second sentences - they contradict each other. It needs to be rephrased.
  4. Lines 323-326: the analogy of previously conducted studies [29] with the study of this work is not legitimately given, since the characteristics of muscle fibers were not studied in the work, conducting research using tasting does not allow such conclusions to be drawn. And in [29], studies were conducted on another muscle group, namely: the gastrocnemius (GAS), soleus (SOL), and extensor digitorum longus (EDL) muscles. The shear force analysis was conducted on a digital tenderness meter. The authors' work does not present instrumental clarifications on the characteristics of muscle fibers. It is necessary to clarify or exclude this conclusion from the discussion.

 Author Response

1.Goose meat contained high values of lead is it recommended to use it for food (Table 4 data)?

It is not recommended to eat meat with more lead than 0.10 mg/kg wet weight according to Swedish Food Agency. The high level of lead is most probably due to direct ingestion of lead pellets or due to those specific birds have been wounded by lead shots. This is most likely only a problem in some individuals and not most geese. With a stricter policy concerning lead ammunition, the problem will hopefully diminish over time. And as we state, further studies are needed on this matter.

2. Is it unclear what the norms are in the footnotes to Table 4? Why are the norms for fish mentioned? Rice cakes?(line 234, 237).

As there are currently no limit values for wildfowl, we included these values as a comparison As we believe that they give perspective to our results. We have included a sentence in the table text to explain this. If you, or the editor, feel that they are irrelevant, we will delete them.

3. Line 246: Is it necessary to specify the methodology for assessing the water-binding capacity? in what kind of meat was determined (boiled or raw?) clarification is required on which characteristics this affects to clarify, or to present in Chapter 5.

Cooking loss was determined as the difference between raw weight after freezing and weight after cooking, now explained in new section 2.2. Lines 136-141.

4. Line 255-257: Since the muscles of the chest (musculus pectoralis major) were used in this study, it is possible that differences will be found in the red (dark) meat of goose and mallard, to which the thigh meat belongs. The muscles of the thigh contain mainly fibers of a rapid type of oxidation (glycolytic fibers), unlike the muscles of the chest. Therefore, the choice of muscle type for research is not entirely clear. The main loads fall on the motor (wing, hip) muscles and the oxidation processes take place in these muscles more intensively. Therefore, it is more expedient to choose red (thigh) muscles for study. This is better suited for assessing the sensory profile, water binding capacity and fatty acid fraction of farm and wild poultry. Therefore, the authors did not find significant differences in the selected muscle type.

We agree that it would have been very interesting to look at other muscles of the birds. However, the reason why breast muscles were chosen was that breast muscles are what is mainly eaten on geese and mallards. The thighs are usually discarded due to not enough meat. Further, in studies on the sensory characteristics of whole muscles in poultry (broilers), it is common to use musculus pectoralis major. We have added a sentence on why breast muscles were chosen over other parts of meat. Another (perhaps peripheral) reason to why breast muscles are interesting to study is that we know that farmed released mallards don’t fly as much as wild mallards, which could be a reason to differences in breast muscle composition. To closer study muscle fiber characteristics would be interesting but does not fit the scope for this journal.

5. Chapter 3.4 (recommended to authors): the above conclusions about the sensory characteristics of meat should be linked to the habitat halo, feeding diets or breed characteristics when discussing the results of this chapter (in the discussion given in lines 284-297, and linked to the data specified in Chapter 3.4)

Exactly how feeding regimes, diets, and morphological differences affect sensory characteristics are hard to answer and was not the focus of this study. However, we do speculate on the matter quite extensively in the beginning of the discussion. We also added text about how a change in feeding regime could change sensory characteristics in lines 315-321.

6. Clarify lines 278 and 281: there is a feeling of contradiction between the conclusions of the first and second sentences - they contradict each other. It needs to be rephrased.

We can see how the text seems contradictory. It is possible to discriminate between a wild female and a farmed female. It is also possible to discriminate between av wild male and a farmed male. It is, however, not possible to discriminate between a wild male and a wild female, or a farmed male and a farmed female. Nevertheless, we have rephrased the sentences to make it clearer. Lines 294-297.

7. Lines 323-326: the analogy of previously conducted studies [29] with the study of this work is not legitimately given, since the characteristics of muscle fibers were not studied in the work, conducting research using tasting does not allow such conclusions to be drawn. And in [29], studies were conducted on another muscle group, namely: the gastrocnemius (GAS), soleus (SOL), and extensor digitorum longus (EDL) muscles. The shear force analysis was conducted on a digital tenderness meter. The authors' work does not present instrumental clarifications on the characteristics of muscle fibers. It is necessary to clarify or exclude this conclusion from the discussion.

We have tried to clarify how, in our study, the characteristics of muscle fibers were only analyzed visually by the analytical panel. We have made changes on line 343-358 and also in the conclusions, line 449-450. We hope this improves the discussion regarding tenderness and muscle fiber composition.

Reviewer 2 Report

The manuscript titled “Evaluation of quality and sensory parameters of meat from  mallard and four species of goose” investigates the nutritional and sensory differences between meat from farm bred goose and mallards, and their wild counterparts. The topic of article is interesting. However, there are some methodological issues that need to be clarified. Therefore, the paper needs Major Revision.

  1. Lines 136-144: Please present details about the analyses.
  2.  Line 148: Instrumental analysis shouldn’t be presented in the paragraph concerning sensory evaluations. Furthermore, please give more details about apparatus that you used and the system of measurement.
  3.     Tables 2-5: Please present results of statistical analyses (p values)
  4.     Table 2: Standard deviations are missing.
  5.   Table 4: Why standard deviation for Pb measurement in goose meat is not present?
  6. Lines 240-253: I can’t find description of methods that you used to obtain presented results. How can you compare raw weights? It’s unclear for me.
  7.   Line 285: The information on heavy metals seems contradictory to you results.

Author Response

1. Lines 136-144: Please present details about the analyses.

A brief overview of the accredited analyses that were used is now added in line 151-154. As the analyses were performed in an accredited, commercial laboratory we unfortunately cannot provide better detail.

2. Line 148: Instrumental analysis shouldn’t be presented in the paragraph concerning sensory evaluations. Furthermore, please give more details about apparatus that you used and the system of measurement.

We agree that this may be confusing. Details about instrumental analyses have been moved to a new paragraph. Lines 136-141.

3. Tables 2-5: Please present results of statistical analyses (p values)

For table 2, statistical analyses could only be performed on iron content in mallards (due to pooling of samples in other groups). Significant differences are now indicated with uppercase letters. This is now explained in the table text.

For table 3, no statistical analyses were possible due to pooling of samples.

For table 4, no significant differences were found. This is now explained in the table text.

For table 5, Significant differences are now indicated with uppercase letters. This is now explained in the table text.

4. Table 2: Standard deviations are missing.

As samples were pooled (consisting of meat from six individuals, lines 147-150), no deviation could be obtained, except for iron content in mallards where standard deviation now is included.

5. Table 4: Why standard deviation for Pb measurement in goose meat is not present?

Same as above (however heavy metals for mallards were analyzed individually, lines 147-150)

6. Lines 240-253: I can’t find description of methods that you used to obtain presented results. How can you compare raw weights? It’s unclear for me.

Details about raw weight, cooking weight and cooking loss were previously included in the Sensory analyses section. After your comments above, it is now included in the new section 2.2. Instrumental analyses, instead.

7. Line 298: The information on heavy metals seems contradictory to you results.

Heavy metals did not differ between types of mallards, thank you for pointing out that mistake. We have now excluded heavy metals in that sentence in the discussion (Line 301).